# Mixed Dye Removal Efficiency of Electrospun Polyacrylonitrile–Graphene Oxide Composite Membranes

**DOI:** 10.3390/polym12092009

**Published:** 2020-09-03

**Authors:** Wongi Jang, Jaehan Yun, Younggee Seo, Hongsik Byun, Jian Hou, Jun-Hyun Kim

**Affiliations:** 1Department of Chemistry, Illinois State University, Normal, IL 61790-4160, USA; wjang12@ilstu.edu (W.J.); jyun12@ilstu.edu (J.Y.); 2Department of Chemical Engineering, Keimyung University, Daegu 42601, Korea; ygseo93@naver.com (Y.S.); hsbyun@kmu.ac.kr (H.B.)

**Keywords:** graphene oxide, reduced graphene oxide, polyacrylonitrile, nanofiber membrane, water purification

## Abstract

Exfoliated graphene oxide (GO) was reliably modified with a cetyltrimethylammonium chloride (CTAC) surfactant to greatly improve the dispersity of the GO in a polyacrylonitrile (PAN) polymer precursor solution. Subsequent electrospinning of the mixture readily resulted in the formation of GO–PAN composite nanofibers containing up to 30 wt % of GO as a filler without notable defects. The absence of common electrospinning problems associated with clogging and phase separation indicated the systematic and uniform integration of the GO within the PAN nanofibers beyond the typical limits. After thoroughly examining the formation and maximum loading efficiency of the modified GO in the PAN nanofibers, the resulting composite nanofibers were thermally treated to form membrane-type sheets. The wettability and pore properties of the composite membranes were notably improved with respect to the pristine PAN nanofiber membrane, possibly due to the reinforcing filler effect. In addition, the more GO loaded into the PAN nanofiber membranes, the higher the removal ability of the methylene blue (MB) and methyl red (MR) dyes in the aqueous system. The adsorption kinetics of a mixed dye solution were also monitored to understand how these MB and MR dyes interact differently with the composite nanofiber membranes. The simple surface modification of the fillers greatly facilitated the integration efficiency and improved the ability to control the overall physical properties of the nanofiber-based membranes, which highly impacted the removal performance of various dyes from water.

## 1. Introduction

Polymer nanofibers have been extensively fabricated to serve as novel membranes for water filtration and purification because of their large surface areas and tunable physical properties, including pore size, porosity, and wettability [1,2,3,4]. These nanofiber-driven membranes are typically prepared by the electrospinning of a polymer precursor solution under a high electrical field. Although the polymer-dependent optimization of the electrospinning conditions (e.g., voltage, humidity, tip-to-collector distance, and viscosity, etc.) is required via trial and error, the overall physical properties can be readily controlled by the diameter and thickness of each nanofiber strand. Bare polymer nanofiber-based materials, however, exhibit inherently weak chemical and mechanical properties for practical applications [5,6,7]. As such, various fillers have been introduced into nanofiber matrices to potentially improve their physicochemical properties for industrial usage [6,8,9,10].

Recently, hydrophilic polyacrylonitrile (PAN) was electrospun to form nanofiber-based membranes for water purification because of its easy handling and well-established electrospinning conditions to regulate the overall properties of membranes [6,8,11,12,13,14]. The proper incorporation of filler materials into PAN nanofibers can possibly induce the reinforcing effect to further improve the chemical and physical properties. However, the use of fillers also requires the extensive optimization of the electrospinning conditions via trial and error to avoid the destruction of their original structures due to their immiscibility [1,2]. Among many filler materials [8,15,16,17,18,19,20], graphene oxide (GO) is of particular interest because it is inexpensive and possesses various physical properties, as well as easy surface functionalization. The integration of GO into polymeric materials has several potential advantages, including promoting an increase in the mechanical strength, permeability, selectivity, and antifouling properties [10,21,22,23,24,25,26]. Preparing composite nanofibers with enough GO to investigate additional new features without significantly disturbing their diameter and shape is an ongoing challenge [7,9,27]. For example, the wettability and chemical stability of the resulting composite nanofibers can be easily regulated as a function of the filler amount. As such, many recent studies have focused on developing a strategy to design GO-based composite nanofiber membranes to utilize their numerous advantages in water purification systems [5,9,11,28,29].

In this work, exfoliated GO surfaces with readily available functional groups (e.g., –COOH, –OH, and epoxy groups) were modified with a positively charged surfactant, cetyltrimethylammonium chloride (CTAC). This simple treatment allowed for the CTAC-modified GO (cGO) to become more compatible with the polymer precursor (i.e., PAN) solution for electrospinning. The systematic loading of GO into PAN nanofibers was then reliably accomplished under the same electrospinning conditions. Although the integration of GO fillers of over 10 wt % into nanofibers is still difficult even under optimized conditions [6,9,24], our approach readily allowed for the loading of GO beyond this limit into the composite nanofibers without causing notable defects. Subsequent heat and pressure treatment of these GO–PAN (cGO–PAN) composite nanofibers turned them into membrane-type sheets with enhanced physical properties. The structural and physical properties associated with the membranes (e.g., nanofiber diameter, wettability, pore size/porosity, tensile strength, and water flux) were thoroughly examined as a function of cGO loading. These cGO–PAN composite membranes exhibited an efficient removal of dyes (e.g., pure dye and mixed dyes), and their adsorption kinetics were examined in an aqueous solution for possible use in water purification systems (e.g., the decontamination of water). The introduction of GO well beyond the typical limits into PAN nanofiber membranes without the significant destruction of the original nanofiber structures can allow for understanding how the loading of fillers into composite nanofiber membranes at high concentrations impacts not only the overall physical properties, but also the adsorption-based removal efficiency to develop complete water purification systems.

## 2. Experimental Section

### 2.1. Materials

*N*,*N*′-dimethyl formamide (DMF) (≥99%) and sulfuric acid (≥98%) were obtained from Duksan Chemical Co. (Gyeonggi-do, Korea). Hydrogen peroxide (H_2_O_2_, 30%) was purchased from Junsei Chemical Co. (Tokyo, Japan). Graphite flakes (SP-1) were obtained from Bay Carbon Inc. (Bay City, MI, USA). Potassium permanganate (KMnO_4_, ≥99.9%) and polyacrylonitrile (PAN, MW = 150,000 Da) were from Sigma-Aldrich (St. Louis, MO, USA). Methylene blue (MB, 95%) and methyl red (MR, ACS reagent) were purchased from Fisher Scientific (Waltham, MA, USA). All chemicals were used as received without further purification. Pure water was obtained using a Millipore system (~18 MΩ·cm).

### 2.2. Preparation of cGO–PAN Composite Nanofibers

Graphene oxide (GO) was initially prepared by our developed approach following the Hummers oxidation and exfoliation of graphite shown in Appendix A [6,30]. The graphite flakes (3.0 g) were stirred in H_2_SO_4_ (77.0 mL) for 1 h at room temperature, followed by the addition of KMnO_4_ (9.0 g). The mixture was placed into a preheated water bath (60 °C) for 1 h. After carefully adding pure water (100 mL) into the mixture, the reaction continued for another 1 h at 90 °C and then cooled to room temperature. H_2_O_2_ (15.0 mL) was then added to the solution and ultrasonicated for 3 h. The mixture was filtered on a filter paper to collect the multilayers of graphite flakes, which were suspended in a mixture of H_2_SO_4_ (1 mL, 98 *v/v* %), H_2_O_2_ (6 mL, 30 *v/v* %), and water (180 mL) for 30 min. The resulting solution was centrifuged at 4000 rpm for 30 min several times until the pH of the final solution became neutral. The graphite oxide powder was then obtained by drying the final precipitates in a vacuum oven. After suspending and sonicating the graphite oxide powder in water (i.e., 0.5 mg dried graphite oxide/mL of water), the solution was centrifuged at 4000 rpm for 30 min. This supernatant solution was filtered with a 0.45 µm PVDF membrane (Millipore Co., Ltd., Burlington, MA, USA) using a fixed dead-end-cell device under pressure at room temperature. A thin layer of GO sheet was then formed on the PVDF filter.

The modification of GO was then completed by mixing GO (0.05 g) and cetyltrimethylammonium chloride (CTAC) surfactant (0.01 g) in 100 mL water by sonication for 2 h and additional stirring for 24 h. After the removal of unbound CTAC by centrifugation (4000 rpm × 30 min × 2 times), the mixture solution was filtered through a dead-end cell to form a thin layer of CTAC-modified GO (cGO) sheets, followed by drying in a vacuum oven. Subsequently, varying amounts of the modified GO (2–30 wt %) were suspended in DMF by sonication, followed by the addition of PAN (MW = 150 kDa) powder. The homogenous mixture was electrospun for 6 h under the following conditions: a voltage of 15 kV, an ejection speed of 0.8 mL/h, and a tip-to-collector distance of 15 cm. The collected nanofiber mats were converted to membrane-type sheets by a heat-press treatment (heating press DHP-2, Dae Heung Sci., Incheon, Republic of Korea) to improve the overall physical properties for water purification [6,10].

### 2.3. Removal of Dyes Using the Composite Nanofiber Membranes

Initially, a stock solution containing 50 ppm of MB or MR was prepared in a 100 mL volumetric flask (Appendix A). The standard solutions of each dye were prepared by a serial dilution of each stock solution. Similarly, a mixed stock solution of MB and MR dyes and its standards were prepared in pure water. These standard solutions were then subjected to absorption measurements using a UV–Vis spectrophotometer (Agilent, Santa Clara, CA, USA) to create calibration curves. A small piece of the cGO–PAN composite (1 × 1 cm^2^) and bare PAN nanofiber membranes was immersed in 3.0 mL of aqueous dye solution (5 ppm). An aliquot of the solution (100 μL) was taken for UV–Vis analysis as a function of time.

### 2.4. Characterization

The structural features of cGO–PAN composite nanofiber membranes were examined by scanning electron microscope (SEM, JSM5410, JEOL Ltd, Tokyo, Japan) after coating with a thin gold layer. Fourier transform infrared spectroscopy (FT-IR, FT/IR-62 spectroscope from JASCO, Tokyo, Japan) spectra of GO and composite membranes were obtained in the scan range of 4000 to 400 cm^−1^. Given the porous nature of the materials, a piece of nanofiber sample (e.g., thin layer) was directly placed into the beam path for transmittance measurements, which were converted to absorbance. Surface enhanced Raman scattering (SERS)_measurements were then carried out using a bench-top ProRaman-L Analyzer (with a spot size of 100 μm and 0.5 NA, Enwave Optronics, Irvine, CA, USA) equipped with a 785 nm laser. The average SERS spectra were collected using a 5 s acquisition time at 5.5 mW of the laser intensity, where the power of the laser source was adjusted with a PM100USB power meter (Thor Labs, Newton, NJ, USA). The thickness of the membranes was estimated by a digital Vernier caliper with a resolution of 1 µm (ABS Digimatic Thickness Gauge, Mitutoyo Corp., Kawasaki, Japan). The pore diameters of the membranes were analyzed with a capillary porometer (Porolux 1000, IB-FT Inc., Berlin, Germany) under wet and dry conditions using a Porewick standard solution with a 16.0 dynes/cm surface tension. The porosity of the samples (4 cm × 4 cm) was examined by measuring the dry and wet weights of the composite membranes after soaking in *n*-butanol for 1 h. The wettability of composite membranes was examined with a contact angle analyzer (Phoenix 300, SEO Inc., Gyeonggi-do, Korea) using a water droplet. A UV–Vis spectrophotometer was employed to examine the removal rate of the dyes over the wavelength range of 190 to 1000 nm. The absorbance peaks were compared to those of the standard solutions of dyes via the Beer−Lambert law. Except for the dye adsorption kinetics, all experiments were performed using three independently prepared samples and the results were collected from the average of at least three measurements.

## 3. Results and Discussion

### 3.1. Structural and Compositional Properties of Composite Nanofiber Membranes

Figure 1 shows the SEM images of various composite nanofibers as a function of the modified GO content. The average diameter and distribution of the individual nanofibers gradually increased upon the addition of GO under the same electrospinning conditions. The composite nanofibers with the GO amount over 10 wt % appeared to be rougher than those with the lower GO amount, but did not show any detectable defects (e.g., beads and nodes). It is noted that the integration of bare GO into PAN typically reaches around 5 wt % [19,31,32,33,34], whereas only a few studies describe the loading of GO up to 15 wt % into PAN nanofibers that required some chemical modifications and the use of additives [9,24]. Unlike polymer matrix modification, stabilizing the guest GO with CTAC via electrostatic interactions greatly increased its miscibility with PAN polymers in a DMF solution to prevent the clogging and/or phase separation during 6 h of electrospinning. This precursor solution with the compatible phase of cGO and PAN resulted in the good dispersity of the GO throughout the PAN nanofibers, which possibly allowed for the loading of GO beyond the typical limits. In addition, our simple strategy did not require extensive optimization of the electrospinning conditions that have often been established by a time-consuming trial and error approach [1,2]. This surface modification could also be applicable to other types of guest substances to prepare diverse composite nanofibers without much alteration of the electrospinning conditions.

FT-IR and Raman spectra of the cGO, bare PAN, and a series of cGO–PAN composites are shown in Figure 2 and Appendix A. Bare GO exhibited a few broad peaks at ~3200 cm^−1^ (–OH), ~1780 cm^−1^ (C=O), ~1630 cm^−1^ (C=C aromatic), 1380 cm^−1^ (C–OH), and 1060 cm^−1^ (C–O–C in epoxy group) [6,22,35]. The CTAC-modified GO exhibited several strong vibrational peaks at ~3200 cm^−1^ (–OH), ~1650 cm^−1^ (a broad band for C=O and C=C), and weak peaks at ~2916 cm^−1^ (C–H), ~1226 cm^−1^ (C–O), and ~1040 cm^−1^ (C–O–C) [36,37], whereas the bare PAN nanofibers displayed distinctive peaks at ~2920 cm^−1^ and ~1450 cm^−1^ (bending of C–H) and ~2240 cm^−1^ (C≡N) [6,7,9,22]. The composite nanofibers showed the detectable characteristic peaks of the combination of cGO and PAN, apparently indicating the effective integration of the GO into the PAN nanofibers. All vibrational peaks with almost identical patterns implied the physical incorporation of cGO into the PAN nanofibers. The Raman spectra also presented the characteristic peaks for all of the GO-containing PAN nanofiber membranes (G-band at ~1610 cm^−1^ and D-band at ~1372 cm^−1^), whereas bare PAN did not show any distinctive peaks. It was reported that the G-band corresponds to the first-order scattering for sp2 carbon domains, and the D-band is associated with disordered graphites, such as defects and edges (i.e., sp3 carbon), indicating a broken symmetry [38,39,40]. The background subtracted Raman spectra shown in Appendix A confirmed the successful preparation of GO, cGO, and cGO–PAN composite nanofibers by examining the ratio of D- and G-bands. While the D/G peak ratio for the bare GO was examined to be 0.88, which was within the literature values [41,42,43,44], the ratio slightly increased to 0.92 and 0.93 upon modification with CTAC and PAN, respectively. It has been reported that the modification of GO (e.g., covalent organic/polymer modifier and chemical doping) often leads to a notable increase of the D-band (i.e., more defects and disordered structure), resulting in the increase of the D/G ratio [43,44]. The marginal increase of D/G ratios for our cGO and cGO–PAN composite nanofibers could be due to the relatively weak modification between the GO and the modifiers via electrostatic and physical interactions. It is also noted that the intensity of these two bands gradually increased without detectable changes in the D/G ratios as a function of the GO content. As such, these two characterizations by vibrational spectroscopy clearly supported the systematic integration of GO into the PAN nanofibers.

### 3.2. Wettability and Pore Properties of Composite Nanofiber Membranes

Figure 3 shows the digital photos and wettability (i.e., water contact angles) of the bare PAN, cGO, and cGO–PAN composite membranes as a function of the cGO content. As we mentioned in the experimental section, the simple surface treatment of GO with CTAC clearly improved the electrospinning process, allowing us to obtain the series of composite PAN nanofibers containing well beyond the typical amount of GO without any problems, including needle clogging and notable bead formation. The uniform color of the composite membranes gradually changed to dark gray throughout the entire membrane with the increase in the cGO amount, implying the successful incorporation of the modified cGO up to 30 wt % into the nanofibers. Subsequently, these PAN-derived nanofiber mats were thermally treated (i.e., 8 layers, 6000 psi at 40 °C) after folding three times to form membrane-type sheets. This thermal treatment was important for improving their mechanical property for use in water purification systems, but the resulting membranes had a greatly decreased hydrophilicity (i.e., from an unmeasurable water contact angle up to 42°) due to the dehydration of the surfaces of PAN nanofibers [6]. Upon the incorporation of the cGO, the cGO–PAN composite nanofiber membranes slowly recovered their hydrophilicity (i.e., decreasing water contact angles) as a function of the GO content. This water wettability trend strongly indicated the successful integration of GO as a filler in the PAN nanofibers where the final cGO–PAN composite membranes exhibited slightly improved hydrophilicity [18,21,45].

In addition, the membrane properties such as bubble point, pore diameter, porosity, and thickness were examined for their possible use in water purification systems (Table 1). The bare PAN membrane possessed a bubble point of ~508 nm and an average pore size of ~306 nm. The bubble point of the cGO–PAN composite membranes initially decreased and gradually increased as a function of the cGO content, while the average pore size was found to have the reverse trend. The significant variation in the pore size between the biggest pore and smallest pore greatly increased with the incorporation of the cGO, possibly due to the structural irregularity of the nanofibers (e.g., diameter and distribution). In addition, the thickness of the composite membranes slightly increased with the integrated amount of cGO. As the cGO–PAN composite nanofiber membranes were prepared under the same electrospinning approach, these property changes were solely attributed to the content of cGO. Unlike the typical GO loading into the PAN nanofibers of less than ~10 wt %, the surface modification of GO easily led to the integration of up to 30 wt % GO. As this preparation approach avoids common electrospinning problems (e.g., clogging, phase separation, and segregation), we speculate that this method can be utilized to design other types of composite nanofiber membranes requiring a large amount of guest materials.

### 3.3. Dye Removal Efficiency of Composite Nanofiber Membranes

To utilize these membranes in water purification systems, the removal of organic compounds was tested using two common dyes, namely methylene blue (MB) and methyl red (MR). Initially, a series of each dye solution was prepared as standards to construct calibration curves by UV–Vis spectroscopy (λ_max_ at 524 nm for MR and λ_max_ at 664 nm for MB, as shown in Figure 4). The dye removal efficiency of the composite PAN membranes was then evaluated as a function of the cGO content using 5 ppm MB and MR solution (Figure 5). Interestingly, the membrane with the 6 wt % cGO (i.e., cGO06 PAN) had a detectably lower removal capacity than other cGOPAN composite membranes. As we do not currently have clear explanations for this observation, a more in-depth study is underway using the composite membranes with the GO content near 5–8%. However, the composite membranes generally improved the removal efficiencies of both dyes via physical adsorption with the increase in the GO content from 0 to 20 wt %. This result clearly revealed the important role of the integrated cGO in the PAN nanofibers. It was reported that various oxygenated functional groups and phenyl backbones of GO can induce attractive forces (e.g., electrostatic attraction, hydrogen bonding, and π–π interaction) to the MB and MR dye molecules [46,47,48]. Thus, the highest dye removal was observed with the composite membranes loaded with the 20 wt % cGO (whereas 30 wt % loaded cGO was too fragile to handle for this water treatment). The presence of abundant functional groups of the evenly embedded cGO into the PAN nanofibers is speculated to induce the enhanced adsorption of the dyes. It is noted that the removal of MR was much higher than MB with all composite nanofiber membranes, possibly due to the compatible polarity between the MR dye and membranes (e.g., the solubility of MB in water is much greater than that of MR). In addition, MR with a lower molecular weight is smaller in size than MB, which could possibly induce greater diffusion through the PAN nanofibers to interact with the abundantly integrated GO [49,50].

The composite membrane (cGO20 PAN) was also tested in the removal of mixed dyes in an aqueous solution (Figure 6). Calibration curves were re-obtained using standard solutions containing two mixed dyes of MB and MR to avoid any matrix-associated absorption errors [51,52,53]. Indeed, the absorbance of the dye mixture was detectably higher than that of the individual dye solution due to the matrix effect [54,55]. The composite nanofiber membranes were then treated in the mixed dye solution as a function of time. The color of the dye solutions changed from purple to light green before and after treatment with the composite membranes, and these colors distinctively differ from the color of the individual dye solutions. Based on the adsorption kinetics using the cGO–PAN composite nanofiber membranes, we observed that the removal of MR was selectively faster than that of MB at a given time interval. The total amount of MR removed was slightly higher in the mixed system when compared to the single MR solution. However, the removal rate for MB was somewhat slower and the total amount of MB removed slightly decreased in the mixed system. This observation evidently indicates the importance of the miscibility between the membranes and dyes. In addition, the adsorption of very polar dyes (homogeneously distributed due to the higher dispersion forces) onto the cGO–PAN composite nanofiber membranes is much slower than that of slightly less polar dyes (non-homogeneously distributed) in water. Since water was used as the main solvent, the mass transfer of evenly distributed hydrophilic dyes (e.g., diffusion) takes much longer to the nanofibrous membranes. Additional driving forces could be causing the dissimilar degree of capillary condensation and/or pore filling [56].

Moreover, four conventional kinetic models were employed using the adsorption process to examine the rate constant (Figure 7) [57,58,59,60,61]. The kinetic equations were used as follows:
$$\text{pseudo-first-order}:\;ln\left(q_{e}-q_{t}\right)=lnq_{e}-k_{1}t$$
$$\text{pseudo-second-order}:\;\frac{t}{q_{t}}=\frac{1}{k_{2}q_{e}^{2}}+\frac{t}{q_{e}}$$
$$\mathrm{interparticle}\;\mathrm{diffusion}:\;q_{t}=k_{3}t^{0.5}$$
$$\mathrm{Elovich}:\;q_{t}=\frac{lna_{e}b_{e}}{b_{e}}+\frac{1}{b_{e}}lnt$$
where *q**_t_* is the adsorbed amount at time *t* (mg/g), *q**_e_* is the adsorbed amount at equilibrium (mg/g), *k*_1_ is the first-order rate constant (g/mg·min), *k*_2_ is the second-order rate constant (g/mg·min), *k*_3_ is the interparticle diffusion rate constant (mg/g·min), *a**_e_* is the initial adsorption rate (mg/g·min), and *b**_e_* is the number of sites available for adsorption. Based on the correlation from the fitting plots (Figure 7), each coefficient value was calculated and the values are summarized in Table 2. The R^2^ values from the pseudo-first-order kinetic model were found to have a better fit than the pseudo-second-order kinetic model for both dyes. Similarly, the theoretical adsorption capacities were closer to the experimental adsorption values (*q**_e_* = 3.024 for MR and *q**_e_* = 2.425 for MB) for the pseudo-first-order model [62,63]. Unlike some of the reports in the literature [49,57,61,63,64], the kinetic model is a better fit for the pseudo-first-order equation in our mixed system, implying the adsorption capacity-dependent process rather than the fraction of available adsorption sites. For the intraparticle diffusion model, the regression line does not pass through the origin, which implies the presence of a discrepancy in the rate of the mass transfer of the dyes (e.g., initial vs. final stages of adsorption). This is possibly due to the porous nature of the composite nanofiber membranes, which could cause the initial boundary layer diffusion effects and different mass transfer through the pores in the later time period. The Elovich model explains the chemical nature of the sorption associated with chemisorption rate (*a**_e_*) and surface coverage (*b**_e_*). The regression line shows a good fit, implying the highly heterogeneous adsorption process of MR and MB dyes onto the composite nanofiber membranes. In addition to characterizing the removal rate of the mixed dyes and their adsorption kinetics, examining the removal of other types of impurities including heavy metal ions and toxic organic species will allow for understanding the adsorption capacity of these cGO–PAN composite nanofiber membranes. Furthermore, evaluating the adsorption mechanisms and water flux under various conditions (e.g., temperature and pressure) will provide an understanding for their potential applications in water purification systems.

## 4. Conclusions

The simple modification of GO with CTAC systematically allowed for the integration of GO up to 30 wt % into PAN nanofibers without electrospinning problems. The ability to prepare these composite nanofibers offered the possibility of evaluating their physicochemical features (e.g., structural and surface wettability) and membrane characteristics (e.g., pore size and thickness) as a function of the GO content beyond typical limits. Increasing the loading of the modified GO filler into PAN nanofiber membranes notably improved the hydrophilicity, which is one important factor in water purification applications. The utilization of these composite nanofiber membranes resulted in a much higher adsorption of organic dyes than the bare PAN nanofiber membranes. A high GO content with various oxygenated functional groups made a great contribution toward the removal of the dyes via electrostatic, hydrogen bonding, and π–π interactions. Investigating the removal capability of organic dyes using a series of composite nanofibers as a function of the GO content clearly provided a better understanding of the role of the GO fillers in water decontamination. Along with examining the removal of organic dyes and their adsorption kinetics, utilizing these cGO–PAN composite membranes in the treatment of heavy metal ions and toxic organic species under various conditions will provide an additional insight for water purification. As such, the proper modification of the polymer nanofibers with hydrophilic GO materials as fillers can lead to the development of novel composite membranes for potential water purification systems with optimal performance.

## Figures and Tables

**Figure 1 polymers-12-02009-f001:**
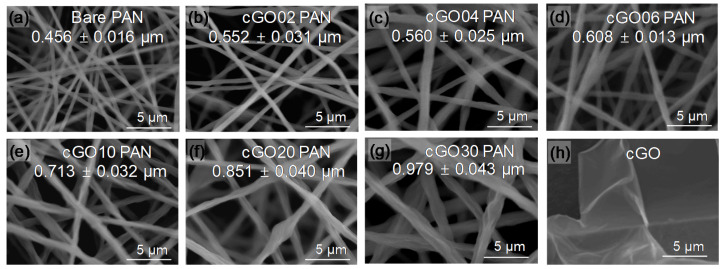
SEM images of (**a**) bare polyacrylonitrile (PAN), (**b**–**g**) cGO–PAN composite nanofiber membranes as a function of cGO content, and (**h**) cetyltrimethylammonium chloride (CTAC)-modified graphene oxide (cGO).

**Figure 2 polymers-12-02009-f002:**
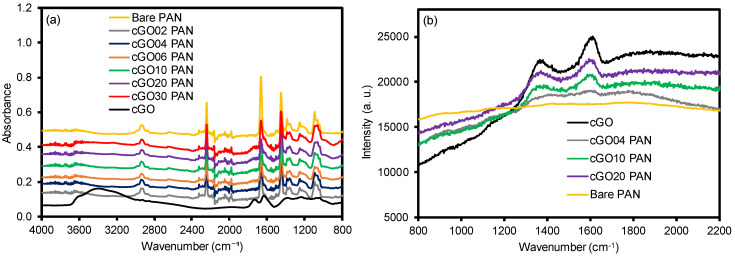
(**a**) FT-IR and (**b**) Raman of bare PAN, CTAC-modified GO (cGO), and composite nanofiber membranes as a function of the cGO content.

**Figure 3 polymers-12-02009-f003:**
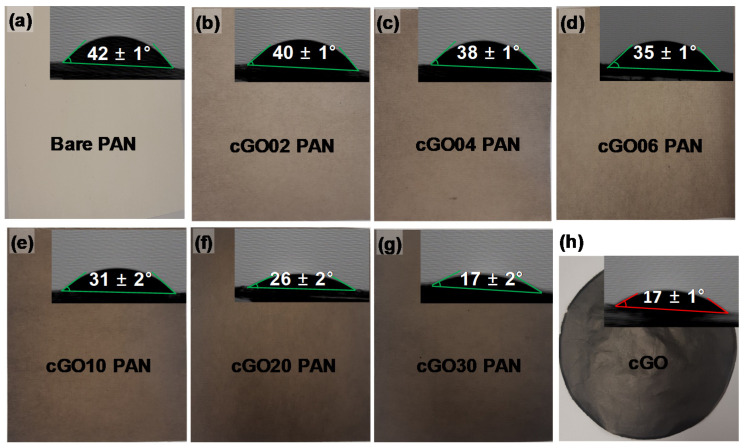
Digital photo and water contact angle of (**a**) bare PAN, (**b**–**g**) composite nanofiber membranes as a function of the cGO content, and (**h**) CTAC-modified GO (cGO).

**Figure 4 polymers-12-02009-f004:**
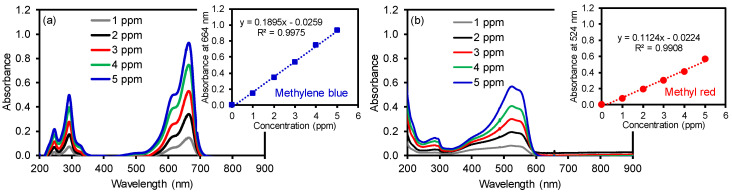
Absorption patterns of (**a**) MB and (**b**) MR as a function of concentration and their corresponding calibration curves.

**Figure 5 polymers-12-02009-f005:**
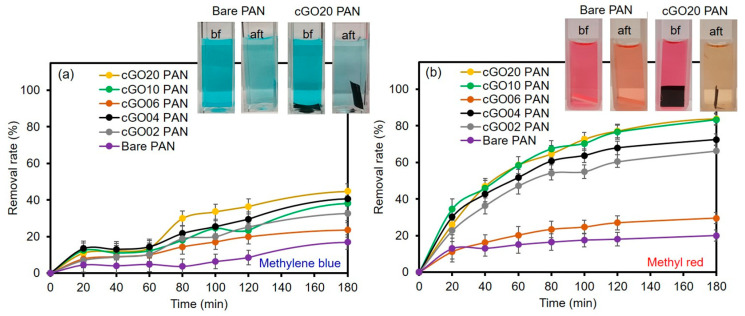
Removal rate of (**a**) MB and (**b**) MR by bare PAN and cGO–PAN composite nanofiber membranes as a function of time (representative color changes of 5 ppm dye solutions shown in digital photos). bf, before; aft, after.

**Figure 6 polymers-12-02009-f006:**
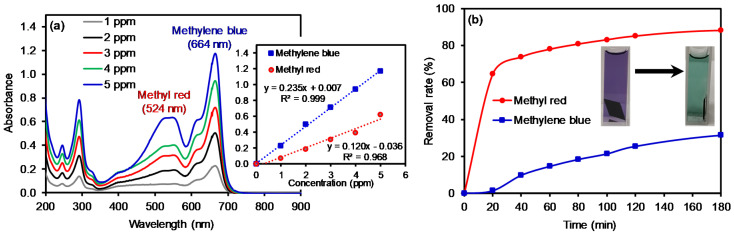
Absorption patterns (**a**) of a mixed solution of MB and MR (5 ppm each) as a function of concentration (inset: corresponding calibration curves) and the removal rate (**b**) of the mixed solution using the cGO20 PAN membrane as a function of time (inset: digital photos of typical color changes of the mixed dye solutions).

**Figure 7 polymers-12-02009-f007:**
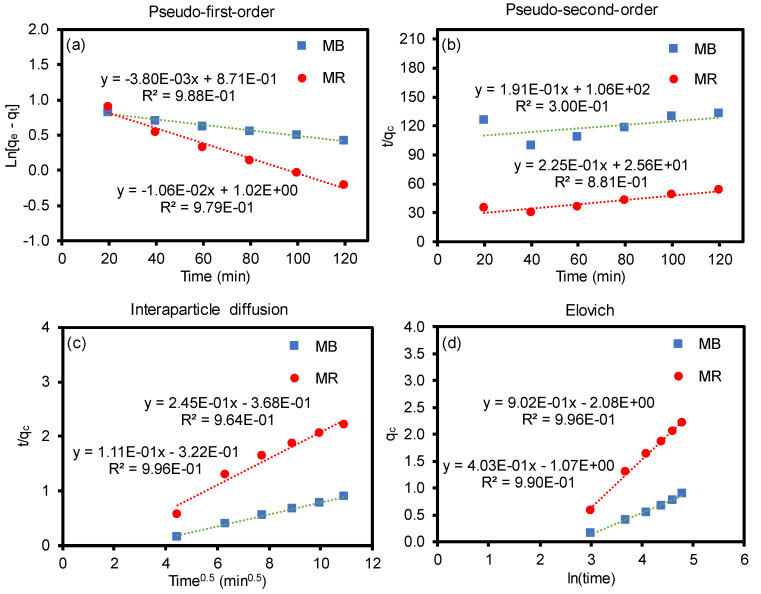
Adsorption kinetic models of MB and MR on the cGO–PAN composite nanofiber membranes (**a**) pseudo-first-order, (**b**) pseudo-second-order, (**c**) interparticle diffusion, and (**d**) Elovich.

**Table 1 polymers-12-02009-t001:** Overall membrane properties of composite nanofiber membranes as a function of the cGO content.

Sample Name	Bubble Point (nm)	Average Pore Size (nm)	Smallest Pore Size (nm)	Thickness (µm)
Bare PAN	507.5	306.0	283.7	90–94
cGO02 PAN	452.7	302.2	244.3	91–99
cGO04 PAN	460.4	319.0	290.2	99–104
cGO06 PAN	494.7	325.8	319.5	93–99
cGO10 PAN	521.6	354.9	306.2	87–93
cGO20 PAN	882.3	253.3	222.1	93–95
cGO30 PAN	966.7	258.3	208.5	102–104

Experiment parameters: 0.715 shape factor and 19.5 mm sample diameter.

**Table 2 polymers-12-02009-t002:** Adsorption kinetic parameters of MB and MR on the cGO–PAN composite nanofiber membranes obtained from graphs in Figure 7.

Kinetic Model and Parameters	First-Order			Second-Order		Diffusion			Elovich		
	*q_e_* (mg/g)	*k*_1_ (min)	R^2^	*q_e_* (mg/g)	*k*_2_ (min)	R^2^	*k* _3_	R^2^	*a_e_* (mg/g·min)	*b_e_* (g/mg)	R^2^
MR	2.773	0.0106	97.9	4.444	0.198	88.1	0.245	69.4	0.0899	1.109	99.6
MB	2.389	0.0038	98.8	5.236	3.442	30.0	0.111	99.6	0.0283	2.481	99.0

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
