# Peer review of "Mixed Dye Removal Efficiency of Electrospun Polyacrylonitrile–Graphene Oxide Composite Membranes"

_polymers, 2020, doi:10.3390/polym12092009_

Round 1

Reviewer 1 Report

In the submitted manuscript, the authors report on the impact of Polyacrylonitrile-Graphene Oxide Composite on the die removal efficiency.

In general, the scalable formation of highly reduced GO and graphene films with improved dispersion in polymers is crucial to the development of graphene-based nanocomposites, and such analyses will significantly push forward the current status-quo for water treatment. The topic studied, the novel synthesis procedure discussed, and the quality of the experiments are suitable, however, it lacks in-depth analysis for the represented area of graphene-based electronics. My main concerns regarding the manuscript are mentioned below:

  1. The introduction is lacking in certain aspects regarding the methods of graphene oxide composites fabrication. These composites can be formed by chemical, thermal (Materials Today21(2), pp.186-192), and other nonequilibrium methods. The nonequilibrium methods are latest discoveries, as they can be employed to inject dopants (like PAN) above the thermodynamic solubility limits due to molten carbon/PAN formation for several nanoseconds, inside the graphene oxide matrix. Such an arrangement can convert the carbon/polymer composite or the polymer itself into graphene oxide or diamond films and polymer composites at room temperature (Materials Research Letters 8 (11), 408-416).
  2. C/O ratios are a deterministic characteristic of GO films, which governs electronic and catalytic activity. The general method of equilibrium-based processing leads to certain stable C/O ratios in rGO films. However, by employing nonequilibrium methods like laser annealing, the GO films with extremely high C/O ratios: as high as 15-16 can be achieved (Carbon2019,153, 663-673) (ACS Appl. Mater. Interfaces 2019, 11, 27, 24318-24330). The authors need to either report C/O ratios in their films by employing the XPS technique or provide representative data from standard Hummers’ processing, thereby comparing the values with other methods mentioned above for the baseline to make sense and make their experiments more comprehensive.
  3. The authors have performed Raman spectroscopy on their specimens. Tuinistra and Koeing methodology is the standard method to establish where in the reduction stage, the GO nanostructures exist. In TK relationship, where initial reduction of GO films results in a notable increment in the ID/ IG ratio. In rGO films, the absence of D′ peak is noted, which is indicative of fewer defect states in comparison with GO. However, reduction beyond a threshold lowers the ID/IG ratio, and the rGO system stops following the TK relationship (ACS Appl. Mater. Interfaces2019, 11, 27, 24318-24330).

The authors need to normalize their Raman spectrum by performing background subtraction to calculate the ID/IG, highlight the presence/absence of D’ peak and compare them with the thermal GO (Materials Today21(2), pp.186-192) and laser processed GO (ACS Appl. Mater. Interfaces 2019, 11, 27, 24318-24330) characteristics to establish that they still have GO in the matrix of the composites.

  1. Further, the authors also need to calculate the defect cluster sizes and their evolution after using Hummers’ method and after the formation of composites by employing the ID/IG formalism. (Nat. Nanotechnol. 2013, 8, 235)
  2. A suggestion would be to go ahead and analyze the dominant contributions towards electron conductivity in these films and the mechanism being followed (ES-VRH or Mott-VRH) and if there is any noticeable change in fitting curves on thermal annealing of these films. (Phys. Rev. B: Condens. Matter Mater. Phys. 2012, 86, 235423)

In summary, the manuscript provides exciting data on the formation and analysis of GO/PAN composites. The investigations are thorough, and the die removal analyses and methodologies are correct. The interpretation of the results is mostly convincing and provides a significant contribution to the area of graphene oxide composites. I recommend the publication of this manuscript. However, the above-mentioned concerns need to be addressed before publication in an archival journal like MDPI Coatings.

Reviewer 2 Report

The manuscript "Mixed Dye Removal Efficiency of electro spun Polyacrylonitrile-Graphene Oxide Composite Membranes" by W. Jang et al. is about PAN nanofiber membranes loaded with GO filler modified with surfactants, wich increases the hydrophilicity and performance of the membrane in water purification applications. Specifically a much higher absorption of organic dyes was observed when GO are loaded compared to bare PAN membranes. A series of techniques was use for the characterisation of the membranes and determination of their efficiency depending on the GO content. This is an interesting subject with applicational potential. I recommend that the manuscript can be published in Polymers, however I would have some comments: 1. why cGO06PAN membrane has lower efficiency than all other membranes, even than the bare PAN membrane for MR, as shown in Fig. 5? Can the authors comment on this and relate it somehow to the morphology (Fig. 1)? 2. which is the composition used for the test reported in Fig.6 and Fig.7? What about a different composition? 3. line 299 - Fig6 - it should be fFig. 7; line 277 - it should be figure 7. 4. Figures 5 and 6 - what is the meaning of the lines joining the points?  

Reviewer 3 Report

The paper reports the preparation of GO-PAN composite membranes and their use for the removal of dyes from water. The preparation methodology is applicable for the preparation of composites membrane with other kinds of fillers. Some interesting results are shown, but some of them are not effectively and suitably discussed, leaving place to some doubts. Major revision is required before publication. I suggest the following changes in order to improve the clarity of the paper:

Major changes:

  • Please, add a figure with the chemical structures of the two dyes (in the manuscript or as supporting information).

  • Fig.3 – please, separate figure 3 and Table. Provide a number and a caption to the Table.
  • Lines 226-227 - Why isn't the removal rate proportionally increasing with the GO concentration? For example, for MB: Removal rate (RR) for cGO 6% < cGO 2%< cGO 10%< cGO 4%< cGO 20%. For MR the bare PAN removes even more than cGO06PAN. Please explain/comment the results.
  • Lines 232-234: The Table shows that cGO20PAN pore size is smaller than the pore size of all the other membranes. Following your hypotesis that the adsorption and diffusion of the dyes are favoured for smaller size dyes, one would expect that adsorption and diffusion should also be favoured for larger pores. The latter situation does not occur. Please, comment on this.
  • For the results shown in Fig.5: which is the initial concentration value for MB and MR?
  • Lines 243-244 and Fig.6 - Only one composite membrane is tested. Which one? Do the authors find similar results for all the membranes?
  • Line 245 - please check: Refs.42 and 44 do not deal with mixed dyes. Ref.43 instead reports that no matrix effect is found. Could you add some references showing such matrix effects?
  • Fig.6 - Specify which is the membrane used here. Which is the initial concentration used for the experiment?
  • Lines 291-296 - I would move this part to the conclusions paragraph, as a plan of future work.
  • Lines 299-300 - Which membrane? It is not clear if the models shown here are applied to the mixed dyes experiments or to the single dyes experiments? Please specify. Do you get differences between the two kinds of experiments?
  • Conclusions section - The conclusions are too general and the role of GO is not clearly described (lines 313-315). The main results of the paper should be listed here along with an evaluation of the most important factors affecting the dye removal rate (hydrophilicity, pore size, interactions, ...). Which is the main influencing factor? Furthermore, the difference observed for the removal of single dyes removal and mixed dyes removal should be also recalled in the conclusions.

Minor changes:

  • Line 77-78 - please check the sentence: “their adsorption kinetics in an aqueous solution for possible use in water purification systems”. Some word is missing, I guess.
  • Fig.2 caption - add the letters (a) and (b): “(a) FT-IR and (b) Raman”
  • Line 193 – “regained” should be “increased hydrophilicity” or “provided a hydrophilic character to PAN nanofibers” or something like that.

Reviewer 4 Report

The paper describes the method for preparation of membranes for water purification. The membranes were produced of cGO-PAN nanofiber in heat and pressure treatment process.

Manuscript presents average degree of significance of content. Major revision of the article is recommended.

2. Experimental

2.4. Characterization

In experimental part the authors show absorbance in FT-IR spectra. Please explain is it only conversion of transmittance or the experiments were performed in ATR mode.

3. Results and discussion

i) How could you explain the lack of any peak originating from CTAC in FT-IR spectra (cGO)? There is no evidence for modification of GO with CTAC. 

ii) Some study showed that the thermal treatment at high pressure influence on the graphene oxide structure by reduction of chemical groups. 

Elemental analysis and/or XPS study of bare GO, cGO and cGO thermally treated should be performed for qualify and quantify of oxygen functional group and modification with CTAC.

3.3. Dye Removal Efficiency of Composite Nanofiber Membranes

The authors compare bare PAN membrane with membrane containing both: GO and CTAC. How the presence of CTAC influence on dye removal.

Removal rate of MB and MR by bare PAN and cGO-PAN composite membranes is unclear. Bare PAN membranes hes higher efficiency in methylene red removal than cGO06 PAN and lower than cGO02 PAN. How could you explain that?

How many times the experiments with dye removal were performed. How many pieces of membranes did you used for water purification?

Round 2

Reviewer 1 Report

I am satisfied with author's corrections to previous version of the manuscript. The additional data provided makes the manuscript more rounded and cohesive, making it suitable for publication. 

Author Response

We are glad to hear that our revision met your expectations.

Thank you very much for your feedback to improve our manuscript.

Reviewer 3 Report

The paper has been revised and the authors have answered all my comments.  However, as concerns one of my comments: “Lines 226-227 - Why isn't the removal rate proportionally increasing with the GO concentration? For example, for MB: Removal rate (RR) for cGO 6% < cGO 2%< cGO 10%< cGO 4%< cGO 20%. For MR the bare PAN removes even more than cGO06PAN. Please, explain/comment the results”, I think that they should discuss this point and report their observations in the manuscript too and not only in the response to reviewers.

After performing this change, the paper can be published.

Author Response

Thank you very much for the recommendation.  We have included the following statements in the manuscript.  We greatly appreciate your contributions to improve our manuscript much better.

"Interestingly, the cGO06 PAN membrane was detectably lower removal capacity than other composites.  As we do not currently have clear explanations for this observation, a more in-depth study is underway using the composite membranes with the GO content near 5-8%.  However, the composite membranes generally improved the removal efficiencies of both dyes via physical adsorption with the increase of the GO content from 0 wt% to 20 wt%."

Reviewer 4 Report

Please correct a typo:

  • line 108 "PVdF filter" should be "PVDF filter".

Author Response

Thank you for finding the mistake.

We greatly appreciate your contributions to improve our manuscript much better.